# What We Know About the Brain Structure–Function Relationship

**DOI:** 10.3390/bs8040039

**Published:** 2018-04-18

**Authors:** Karla Batista-García-Ramó, Caridad Ivette Fernández-Verdecia

**Affiliations:** 1Images Processing Group, Basic Division, International Center for Neurological Restoration (Ciren), La Habana 11300, Cuba; 2Biomodels Laboratory, Basic Division, International Center for Neurological Restoration (Ciren), La Habana 11300, Cuba; ivettef@neuro.ciren.cu

**Keywords:** brain, connectivity, structure–function relationship, network theory, computational modeling

## Abstract

How the human brain works is still a question, as is its implication with brain architecture: the non-trivial structure–function relationship. The main hypothesis is that the anatomic architecture conditions, but does not determine, the neural network dynamic. The functional connectivity cannot be explained only considering the anatomical substrate. This involves complex and controversial aspects of the neuroscience field and that the methods and methodologies to obtain structural and functional connectivity are not always rigorously applied. The goal of the present article is to discuss about the progress made to elucidate the structure–function relationship of the Central Nervous System, particularly at the brain level, based on results from human and animal studies. The current novel systems and neuroimaging techniques with high resolutive physio-structural capacity have brought about the development of an integral framework of different structural and morphometric tools such as image processing, computational modeling and graph theory. Different laboratories have contributed with in vivo, in vitro and computational/mathematical models to study the intrinsic neural activity patterns based on anatomical connections. We conclude that multi-modal techniques of neuroimaging are required such as an improvement on methodologies for obtaining structural and functional connectivity. Even though simulations of the intrinsic neural activity based on anatomical connectivity can reproduce much of the observed patterns of empirical functional connectivity, future models should be multifactorial to elucidate multi-scale relationships and to infer disorder mechanisms.

## 1. Introduction

How the human brain works is still an open question, as is its implication with brain architecture: the non-trivial structure–function relationship. The development of neuroimaging techniques, the improvement on their processing methods and the unfolding of computational neuroscience field have driven brain research focused on the relationship between anatomical and functional interactions. The principal hypothesis is that the anatomic architecture determines, but not strictly, the network dynamics [1,2,3,4], meaning that part of functional connectivity cannot be explained by considering only anatomical connectivity. This hypothesis is based not only on human studies but also on animal models, with the advantage that they can be correlated with genetic, histological and molecular methodologies. Network theory, computational modeling and complex system analysis have played pivotal roles in elucidating structure–function relationship.

Studies on connectivity patterns, both functional and anatomical, in humans and animals [5,6,7,8,9,10], and their relationship with neurodegenerative conditions such as Alzheimer’s and Parkinson’s diseases [11,12,13,14,15,16] and multiple sclerosis [17,18] has been published in recent years. The brain is considered the most complex system in the Animal kingdom, with complex space-time patterns, and where the degree of correspondence between structural and functional connectivity depends on spatial resolution and time scales [19]. These elements in turn are continuously modified based on the sensorial inputs that enter the system and the feedback that emerges from the interaction between the processing of these signals and the emergence of eferences whose result is functional and adaptive remodeling at different organizational levels. 

Computational models allow inferring the network dynamics from empirical anatomical connectivity obtained from diffusion weighted imaging. Several authors showed that the similarities between the simulated functional matrix, by implementing different types of computational models, and the empirical functional matrix, obtained from functional neuroimaging techniques, are maximum when the dynamics of the global network operates on a critical point of state transition [20,21,22]. This finding is supported by empirical evidence, from functional neuroimaging techniques, that show that the dynamical regime of resting state corresponds to critical point of state transition [23,24]. Critical dynamics are crucial from the functional point of view given that in this critical state the system presents an optimal sensitivity to possible inputs and a maximization of the number of different functional states available. This allows the system to rapidly compute a specific brain function according to the stimulus received. These findings suggest that this is an optimized state for information transmission and for early adaptation to external perturbation.

However, the high specificity of the mechanisms that condition the structure–function relationship of neural systems is today an incompletely answered question. There is evidence of the modulation of this interrelation by activity and behavior in its broadest sense, but how this feedback works in the time-space context of neural systems is still a big question. This involves complex and controversial aspects of the neurosciences and that the methods and methodologies to obtain both structural and functional connectivity are not always rigorously applied. The heterogeneity regarding methodological approaches and polemic points such as the mechanism driving functional connectivity and spatio-temporal scales make the comprehension and interpretation of the structure–function relationship difficult.

There are questions involving structure–function relationship without a definitive answer: How can the functional activation patterns be changed? How do plasticity mechanisms become involved? In what way and under what conditions do different favorable plastic responses, neuroprotective and successful from the adaptive point of view, happen to be aberrant events and therefore an expression of maladaptation? A deeper understanding of structure–function relationship is still difficult today, as is how this relationship is modified in some neuropathologies. 

There are reviews of this topic from different perspectives [19,25,26,27,28]. We have organized this review summarizing those concepts related to the structure–function relationship within the framework of neuroimaging; that is, updated definitions of anatomic and functional connectivity, the methods used to extract these measures, and their interpretation under different spatial-temporal scales. The following section deals with the network theory and computational modeling related to brain function. Afterwards, we discuss the main findings on the structure–function relationship under neurodiscapacity conditions. Finally, we expose limitations identified in current studies to be considered in future research on this topic.

## 2. Definitions

### 2.1. Structural Connectivity

Anatomical or structural connectivity (SC) refers to physical synaptic connections between neural elements or the white matter fibers connecting gray matter regions, depending on the spatial scale. Tract tracing technology constitutes an invasive method for estimating the weight of axonal connectivity between a pair of pre-defined brain regions at micro-scale. This technique allows a deep characterization of synaptic termination of connections at cellular level and an assessment of the directionality of white matter projections. Tract-tracing studies have allowed collecting connectomes of the mouse [29,30], rat [31] and the macaque [32]. On the other hand, diffusion weighted imaging (DWI), an MRI technique, is a non-invasive method to record in vivo white matter’s architecture [33]. The present review focuses on this non-invasive technique that corresponds to the macro-scale, but its correspondence with the micro-scale is analyzed below. DWI really provides information about the restrictions on the diffusion of water molecules, which is used to infer the orientation of white matter tracts (for review, see [34]). The main limitations of SC based on DWI are that the connections are non-directed because tractography algorithms cannot determine the direction of axonal projections, and they cannot differentiate between excitatory or inhibitory connections. These limitations are due to intrinsic features of neuroimaging technique.

Different tractography algorithms and software packages have been developed to estimate the likelihood of the existence of connections between two regions from DWI data. In general, these methods can be classified into two types considering how the white matter tracts are modeled within the voxel: probabilistic tractography based methods [35,36,37] and deterministic fiber tracking tractography [38,39,40,41]. The streamline approach is advantageous to characterize long white matter tracts. However, the probabilistic tractography is more effective in resolving crossing fibers [42]. In any case, using both methods, it is possible to construct structural networks. Bastiani et al. [43] argued the advantages and disadvantages of tractography algorithms considering not only deterministic or probabilistic distinction but also whether the model is a single-direction or multiple-direction intra-voxel diffusion model and comparing local versus global tractography.

Several measures are used to quantify anatomical connectivity: the probability connection, the fiber count and fiber’s length. Based on these, the density and strength of connection has been defined. These measures have often been used interchangeably in the study of structure–function relationship. There is no consensus on a measure of anatomic connectivity, hampering a comprehensive understanding of the findings obtained by diverse studies. 

### 2.2. Functional Connectivity

Functional magnetic resonance imaging (fMRI), electroencephalography (EEG), magnetoencephalography (MEG) and positron emission tomography (PET) are imaging methods of macroscopic brain activity. Each of these techniques is sensitive to different physiological processes; therefore, they have different spatio-temporal resolutions. For instance, the non-invasive fMRI technique has high spatial resolution (~1 mm) and poor temporal resolution (seconds). In contrast, EEG/MEG measures neuronal activity with excellent temporal resolution and low spatial resolution. Functional connectivity (FC) has been defined as statistical dependencies between distinct and distant regions of information processing neuronal populations [44,45] and can be extracted from anyone of these techniques including the so-called “resting-state” fMRI (rs-fMRI) [46,47]. Regarding structure–function relationship, the activity at slow time scale of the rs-fMRI signals seems more constrained by structural configuration than the faster time scales of the EEG. Functional connections estimated at larger time windows strongly overlap with the underlying structural connections, while, for smaller time windows, there can be a structural–functional network discrepancy due to distributed delays between neuronal populations that cause transient phase (de-)synchronization [1,7]. For this reason, this review focuses on the findings based essentially in fMRI, specifically rs-fMRI. There is still much debate about the neuronal basis of rs-fMRI, although it has been accepted that the rs-fMRI signal is due to the intrinsic activity dynamics that reflects, at least in part, aspects of the functional capacity of neural systems [48]. Various authors established a relationship between time-varying functional resting-state connectivity and the network topology using large-scale models, which is discussed below.

The blood-oxygen-level dependent (BOLD) signal of fMRI is an indirect measure of brain neuronal activity. Although what it really reflects are the differences in blood oxygen level in brain tissue. The methods proposed to study FC from fMRI have been classified in two general categories: model-based methods and data-driven methods. Model-based methods such as general linear model [49], cross-correlation analysis [50] and coherence analysis [51] are based on prior knowledge. In contradistinction to these methods, data-driven methods are independent of prior information; some of these are principal component analysis (PCA), independent component analysis (ICA) and clustering analysis. Each method has own advantages and limitations (for review, see [52]). 

A variety of measures have been used to describe FC: correlation, covariance, coherence, Granger causality, and mutual information. Pearson correlation coefficient between the time series of different regions is the most used functional measure. The interpretation of functional connectivity metrics is a crucial point in the comprehension of the relation between structure and function [53] because the FC may reflect direct interaction between two regions but also may exist between anatomically unconnected regions, in contradistinction to SC that reflect white matter fibers that physically connect two regions. The FC is dealt with as a purely spatial measure and this static description is a moot point given that the non-stationary of resting-state connectivity turns around the way that functional interactions are examined and simulated.

The definitions and some limitations to be considered before starting any connectivity study have been reviewed, but an important point that has not been mentioned in obtaining the anatomical and functional connectivity matrices is the parcellation scheme. In connectivity matrices, the columns and rows (targets and sources) represent the neural elements i andj, and c_ij_ entries represent neural connections. The number of neural elements depends on the parcellation schemes of the brain into regions utilized (for review, see [54]). There are different anatomical brain atlases that are used to define the nodes of the connectivity matrices. The most widely used anatomical atlas is the Automated Anatomical Labeling (AAL) that comprises 116 ROIs (nodes) [55], which uses the parcellation proposed by Hagmann et al. [2]. The various parcellation schemes differ in the number, shape and location of ROIs which can alter connectivity profiles and obstruct the comparison of results across studies. In the particular case of FC, it is likely that they include signals from different functional regions due to the usually large size of the ROIs derived from most parcellation schemes.

### 2.3. Spatio-Temporal Scales

The brain has been described as a complex network on multiple spatial scales, from synaptic circuits to whole-brain system to process and integrate information. Essentially, it is split into three levels: macroscopic, mesoscopic or microscopic scale. At macroscale, we handle with brain regions and white matter fibers connected these regions. The microscopic scale is extended to single neurons, the dendrites, axons and synaptic activity. However, how these scales are related is still an open question. Even though this review focuses on empirical connectivity obtained from neuroimaging techniques, which corresponds to the macroscale, a comprehension of structural correspondence and functional interplay between these two scales is imposed. In this sense, a significant correlation between microscale cellular metrics with macro-cale network properties of macaque cortex has been reported. For instance, highly connected regions show more neural complexity in terms of dendritic branching, soma size and spine density compared with lowly connected regions [56]. A correlation between cytoarchitectonic properties, such as neuron size, and macroscale connectivity, reflecting an association between local and global organizational features of human cortex, has also been found [57]. Likewise, a relationship between the functional connectivity and the underlying chemoarchitecturehas been shown [10,58,59]. Even though there is evidence of the correspondence between micro- and macroscale, it has not been possible to explain the physiological mechanisms of many of the phenomena experimentally observable at macroscale. Thus, a third scale has been defined, the mesoscopic level, which represents the transitional state between the macro- and microscales. Mesoscopic refers to neuronal populations under the assumption that the neurons of a population have similar properties with the aim of explaining macroscopic spatio-temporal dynamics from mesoscopic approximations.

The brain has been studied essentially in two time-scales: short-term (seconds to minutes) and long-term (days to years) time-scales. The structural configuration is relatively stable across time but can change due to the development and neuroplastic processes, such that the SC (at macroscale) remain stable on short-term time scale while neuroplasticity can be observed on long-term time scale. On the other hand, we have already mentioned that FC is not time-invariant; relationships between neural elements can quickly vary across time. FC from fMRI studies corresponds to macroscopic scale due to the spatial resolution that is not enough to directly represent the neuronal dynamic activity at microscopic level.

## 3. What We Know About Structure–Function Relationship

Neuroimage processing, network theory and computational modeling have played essential roles in the study of structure–function interactions. Empirical studies obtained the structural and functional matrices where nodes represent regions and the edges represent the connections between these regions (connections can be anatomical measurements or functional correlations), and the correlation between them, structural and functional connectivity, is calculated to evaluate the closeness of their relationships. In the cases where the SC derived from the DWI studies and the FC derived from the rs-fMRI of the same subjects have been compared, different correlation coefficients between functional and structural connectivity matrices have been reported in approximately a range of 0.4–0.7. However, these studies use different methodologies, namely different neuroimaging processing methods, different subdivision schemes and different connectivity measures.

Some papers implement computational models to infer the FC from SC obtained from DWI, and compare the inferred FC with empirical FC obtained with functional neuroimaging techniques, e.g., rs-fMRI or EEG. A consistent result of these studies of the structure–function relationship is that resting-state network reflects at least in part the underlying SC. This has also been demonstrated in mice [60]. This relationship between structural network and network of dynamic couplings has been demonstrated at macroscale [3,40], mesoscale [61], and microscale [62]. However, part of the FC is not supported by the underlying SC and this may be due to several factors. For example, the FC between two regions can be given in the absence of a structural connection between these regions, that is, it can be mediated by third areas. On the other hand, another factor that influences the non-total correspondence between SC and FC is related to the dynamic aspects of the FC, that is, the non-stationary nature of FC. A dependence on the SC–FC relationship with the time scale considered in the study has been described [1,7]. Mišić et al. [6] illustrates that, at network-level, a particular structural configuration supports a different non-overlapping functional configuration, which means that functional network does not necessarily correspond node-to-node to the underlying structural network.

The brain has been studied using network analysis and a common architectural characteristic has been found in both anatomical and functional networks. A full description and interpretation of graph-theoretical approach is beyond the scope of this review (for review, see [63]). A group of measures that reflect the integration and segregation processes allow us to characterize structural and functional organization of the brain, but interpretation of these metrics depends on the choice of nodes and edges, again the parcellation problem (to define node).

Even when different experimental methodologies and different parcellation schemes were used, there is evidence that the brain has small-world properties, which means a tendency to clustering nodes into modules reflecting an efficient topological integration and economic wiring, for both functional [64,65] and structural networks [2,40,64,66,67]. This “small-world” network property has been associated with optimal communication efficiency and high-speed information transmission due to the coexistence of high clustering and short average distance, which facilitates the integration and spreading of signals [68]. Despite all this preliminary evidence, the assumptions of the small-worldness have been questioned [69]. Another feature that has been demonstrated in humans is the rich-club organization [70,71], which means a presence of highly interconnected regions (hubs), beyond what is expected considering only the node degree, which represents the number of connections that link the node to the rest of the network. This rich-club organization has been associated with the shape and route that global processes follow in neural system and that contributes to the integration process. Rich-club structure is essential in cross-linking of functional modules in the human brain [72]. These two properties, the modular structure and the rich-club organization, seem to be determinants in the dynamic activity of the brain that is affected when one of these properties is destroyed [9]. Other network measures have been directly related to the brain’s activity. For example, a previous study found that the shortest paths of the structural network were strong predictors for functional connections [68].Some of these large-scale topological features are conserved in some species. Small-world network organization, the modular structure and central rich club organization have been shown in rat [31], mouse [60], cat [73] and macaque monkey [1,74,75]. These studies allow us to conclude that there are similar topological organizational features of neural network architecture among species.

As mentioned before, the relationship between SC and FC does not exhibit a simple one-to-one mapping, thus modular structure helps clarify this relationship. Diez et al. [76] proposed a brain partition based on structural–functional modules. They hypothesized that there is a brain partition common to both structure and function networks. They found a strong correspondence between brain structure and resting state dynamics by implementing a standard hierarchical agglomerative clustering algorithm. Their work confirmed that both rs-FC and SC networks display a high modularity, and that there is an excellent matching between functional and structural modules.

In addition to the graph-based approaches that have helped to clarify structure–function relationship, important contributions have also been made by computational modeling. In the last decade, many mathematical and computational models have been proposed that can generate detailed neuronal dynamics to make inferences about brain functionality and its complex behavior. These models have contributed to bridge the gap between anatomy and brain dynamics by making inferences to what extent the anatomical configuration predicts neural dynamics, for instance by comparing simulated functional connectivity with empirical one. A crucial point of computational models is the trade-off between complexity and realism. In general, the large-scale brain network models that incorporate realistic anatomical connectivity to simulate neuronal population activity in each region allow a balance between biophysical realism and model complexity owing to its low-dimensionality. These models are based on mesoscopic approximations: neural-mass and mean-field reduction [77], and a group of neurons (neuronal population) that shares the same physiological properties at a given physical location is only influenced by the mean activity (activity modeled by a set of dynamical equations) of another population. These neural populations can be coupled together according to empirical measures. These mesoscopic approximations allow a close up image to comprehend the underlying physiological mechanisms (microscale) that give rise to spatial-temporal macroscopic dynamics in the healthy and diseased brain. In this context, several works have utilized computational models to generate neural dynamics from the empirical structural connectivity matrix [1,3,7,20,21,22,73,78,79,80]. The general methodology followed by many of these articles consist of: (1) obtain structural and functional connectivity matrices from DWI and rs-fMRI, respectively; (2) simulate activity from structural connectivity matrix using a computational model; (3)estimate BOLD signal from the simulated neural activity; and (4) compare empirical versus simulated FC. Following this general methodology, all these works have elucidated and reinforced key points about the relationship between resting state networks and the underlying anatomical connectivity. First, there is a tendency towards high FC between regions strongly anatomically connected, as well as between regions without direct structural connections. Second, there is a dependence on this relationship of the time scales: st a slow time scale (minutes), the FC reflects part of the underlying structural network, snd not the FC fluctuations that take places on shorter time scales. Third, it is suggested that these non-stationarities could explain the part of FC not supported by the underlying SC. Fourth, the best fit between simulated (using SC matrix) and empirical (measured experimentally using fMRI) FC is when the network state is at the edge of a dynamical bifurcation point. This last finding reflects that there is a functionally significant dynamic repertoire that is inherent of the structural configuration. That the brain network operates at the edge of instability is functionally relevant because it indicates the existence of a set of available brain states that can be activated and stabilized quickly and easily when necessary, for example before a given task or for a given function.

The structure–function relationship has not only been studied in humans, similar results have also been obtained in animal studies. An advantage of animal models is that they allow studies at the cellular and molecular level, on either a micro-or a sub-microscale. A correspondence between anatomical and functional connectivity in somatosensory cortex has been demonstrated for anesthetized monkeys [28] and in mice [5,60]. Díaz-Parra et al. [8] simulated FC using rat connectome derived from tract tracing experiments to compare with empirical FC obtaining from rs-fMRI obtaining a Pearson correlation of 0.53, which intensifies the hypothesis that anatomical topology shapes, at least in part, functional networks. They also found that functional modules are enriched in densely connected anatomical motifs. 

## 4. Structure–Function Relationship in Neurological Disorders

The objective of this section is to cite some examples where the structure–function relationship has helped to elucidate more about neural mechanisms; for example, in the normal aging process and in some pathologies, particularly those that have been described as network disorders such as epilepsy, schizophrenia and autism.

Considering what has been said up to now, the correspondence between SC and FC would be a better biomarker (greater sensitivity) of brain disorders than those biomarkers based only on imaging modalities. Why some functions are preserved after similar structural brain damage in some patients and not in others is an open question. In this framework, inter-individual variability in the evolution, behavior and neurological profile among patients with the same pathology has been associated with different structure–function relationship between patients. That is, more than diseases, there are patients and, therefore, the personalized medicine forms the basis of the future clinical-therapeutic approach.

### 4.1. Aging

Aging is associated with changes in brain morphology as well as with a decline in cognitive performance; in fact, several authors reported anatomical and functional impairment in different brain regions. Persson et al. [81] reported anatomical and functional differences associated with cognitive decline between two groups with different levels of episodic memory performance over time. They found a reduction in anatomical connectivity in the anterior part of the corpus callosum and an increase in the recruitment of frontal regions in the group with a declining memory performance. In fact, they found a negative correlation between anatomical measurement and the increment of the activation pattern (fMRI), in the form of an increase in the number of activated regions during a cognitive task performance. They argued that this result suggests that the additional activation pattern during task performance could be a compensatory mechanism for the structural disruption associated with memory decline. On the other hand, Nakagawa et al. [27] demonstrated, based on computational modeling, that the decrease in structural connectivity in aged people led to lower complexity of BOLD signals. The complexity allows quantifying temporal patterns of BOLD signal; higher complexity of brain activity leads to richer and more integrated information in the network. They argued that it could reflect changes in processing speed which would explain the cognitive performance decline distinctive of aging. Although the results may seem contradictory, they are not, since the first study is based on the activation pattern during the execution of a certain task, while the second is based on “resting-state” fMRI, in the absence of an explicit task. These results seem to indicate that deterioration in the structural network during aging causes a decline to information processing so that the system needs higher overall activation to cognitive performance. These two results are examples of how the structure–function relationship can be used as biomarkers, in this case of aging.

### 4.2. Epilepsy

Studies from epileptic patients have reported alterations in both structural and functional connectivity as well as a negative correlation between both parameters. Liao et al. reported a decrease in both FC and SC in mesial temporal lobe epilepsy (TLE) patients when compared with controls [82]. Zhang and coworkers corroborated these findings in patients suffering idiopathic generalized epilepsy (IGE) [83]. Furthermore, the correlation between SC and FC shows a tendency to decrease in TLE patients vs. controls [84] similar to that described in IGE patients [84]. Nevertheless, all these studies referred to an interesting finding: the SC–FC correlation increased during the epileptic crisis. It could be suggesting that the SC–FC correlation is a possible biomarker for epilepsy monitoring. 

A network topology study in children with frontal lobe epilepsy compared to age-matched controls revealed a relationship between functional networks impairment with an increment of the cluster coefficient and path length of the epileptic patients group. This finding represents the existence of networks with tightly clustered modules but with limited inter-modular connectivity. Because this study did not find significant differences between groups for structural networks, it has been suggested that the functional alteration precedes structural changes [85].

### 4.3. Schizophrenia

Schizophrenia has been described as a disorder of brain connectivity [86,87]. Numerous topological disturbances have been reported on structural and functional networks [86,87,88]. Skudlarski et al. [86] found a diminished SC accompanied by either decrement or increment of FC in different regions (hypo- and hyper-connectivity) when comparing connectivity maps of schizophrenic patient group with control group. Some of these functional abnormalities are supported by anatomical changes; the complex nature of the structure–function interaction is also evident in schizophrenia [88]. Lastly, Stephan et al. [89] reviewed computational models based on neuroimaging data and focused on schizophrenia as a spectrum disease. In cases such as schizophrenia, where a spectrum is described, individual analyses are imposed for each patient (for review, see [87]).

### 4.4. Autism

Autism spectrum disorder (ASD) has also been characterized as a network disorder with abnormal anatomical and functional connectivity. One of the reported findings is a decrease in inter-hemispheric FC [90] that seems to be related to reduced inter-hemispheric anatomic connectivity [91,92]. However, an SC increased in young children [93] and adolescents [94] with ASD has also been reported. Thus, there are questions such as: Is connectivity disorganization part of the primary pathogenesis in ASD? Are there differences in connectivity patterns between distinct syndromes of ASD? Basic and clinical studies will shed light on these issues.

## 5. Limitations

In this review, the most relevant methodologies in neuroimage processing, network topology analysis and computational modeling used to study of structure–function relationship are discussed. However, the images acquisition and processing per se offer inherent limitations of the methodology. For instance, when structural connectivity is based on DTI, it can ignore long-distance and fiber-cross connections and does not provide information about the directionality of the connections [2,42]. In the case of fMRI, it is important to note some aspects: neuronal activity is not directly a measure of the BOLD signal of each voxel; it is an integration of a variety of neuronal activities and the increase of excitatory or inhibitory synaptic activity can cause an increase of metabolic activity. Such limitations, properties of neuroimaging techniques, may lead to imprecise brain network representations, affecting the analysis of network properties such as the study of the structure–function relationship. 

Regarding the methodology followed for image processing, there are two points that limit the interpretation of the results as well as make comparison between the different studies difficult: parcellation schemes and connectivity measures. The definition of nodes and edges is a critical step to obtain connectivity matrices; in fact, the limitations on the application of graph theory and the validity in its interpretation depend on graph representation itself: nodes and edges. Until now, there is no universally accepted parcellation scheme: the diverse anatomical brain atlases exhibit remarkable differences in the number, shape, and location of ROIs. The results of anatomical and functional connectivity depend on the algorithm and parcellation scheme considered. This dependency of the results of anatomical/functional connectivity on parcellation schemes and the non-standardization of connectivity metrics cause inconsistencies between studies. Taylor et al. referred to the use of a high-resolution parcellation scheme (50,000 nodes), focusing the analysis on modularity within and between brain areas [95]. They described a modular architecture within brain areas that could be the structural mechanism to implement different functions.

In addition, the variety of connectivity measures for both structure and function make comparison across studies difficult. Huang and Ding argued that node weight, a measure that considers the number and length of fibers and the ROI size, and conditional Granger causality are more appropriate measures to link functional and structural connectivity [53].

In summary, the extent to which the different methodologies and procedures (connectivity measure and graph construction) faithfully reproduce the biological phenomenon itself is something that requires additional studies to establish guidelines in this regard.

In the case of computational modeling, models with many variables make parameter estimation very complex. In addition, it is not clear to what extent the values of some parameters affect how well models predict FC. Some models do not differentiate neural regions and are based on certain global parameters, coupling weight, transmission speed and noise, which determine the global network activity. All these elements must to be carefully considered when a computational model is used to study SC–FC relationship. 

## 6. Conclusions

From surveying the current literature, what is clear, despite all methodological limitations, is that resting-state functional connectivity is constrained by the large-scale brain anatomical configuration, but other factors also influence it. Multi-modal neuroimaging techniques and improved methodologies are needed to obtain structural and functional connectivity matrices. Another consistent finding is that simulations of intrinsic neural activity based on anatomical connectivity can reproduce much of the patterns observed in empirical functional connectivity. Computational modeling will continue to be an increasingly essential scientific endeavor, playing an increasing role in elucidating multi-scale relationship sand inferring disorder mechanisms.

Clearly, more research is imperative to deepen the understanding of the structure–function relationship. Longitudinal investigations can be fruitful in future works to evaluate neuroplasticity processes. In addition, the wide variability among patients with the neurological disorders imposes personalized individual treatment as health-care options.

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
