# Peer review of "What We Know About the Brain Structure–Function Relationship"

_behavsci, 2018, doi:10.3390/bs8040039_

Round 1

Reviewer 1 Report

Authors have followed most of my advice and the updated manuscript reads better that the previous version. However, I would still like to urge the authors to at least mention the growing evidence against the small-world topology of brain networks (see my previous report).

Overall the English is rather poor throughout the manuscript -- this also true for the newly added text. It is very important that the manuscript is revised by a native speaker of English language. Finally, there are few more point that authors should take into account while revising the manuscript -- which I do not have to review again and editors can make the final decision.

Minor edits:

L60: network's neuronal dynamic -- network dynamics

Line 361-362: This is not any better than the sentence authors had in the previous version. What do you mean by 'some' -- on what basis did you select those exampled. What do you mean by 'mentioned' -- what aspects are you discussion? Please introduce this section better.

Line 378-379: Not clear what authors mean by 'increased recruitment'.

Line 382: define what is complexity of bold signal

Section 4.1: Authors should state how 'increment of activation pattern' relates to 'lower complexity of bold signal'. And in what exact way resting state is a bio-marker? What can we infer from bold in resting state or evoked state? 

Author Response

We appreciate and we are very satisfied with the work done by the referees.

Reviewer 2 Report

Dear Authors, 

If I am understanding this work properly, your main thesis is that functional connectivity in the brain can not be directly inferred from structural connectivity. The lack of concordance is due in part to both the underlying biology and limitations in current neuroimaging modalities, which prevent simultaneous acquisition of both high resolution structural and functional data in the living brain. And you suggest that a structure/function correlation metric could serve as a biomarker for neuropathology. 

The structure/function relationship is a compelling premise for a review. I think the structure of the paper works well. My main suggestion is to be more specific. It would help to know which brain regions you are referring to when discussing the extent to which spatial and functional connectivity correspond. 

Some ideas are underdeveloped. For example, the idea that computational models of functional connectivity perform better around state transitions was mentioned in the introduction (line 58), but not revisited in further sections. Could information from state transitions be used to gain more detailed insight into the structure/function relationship? 

Is is possible to acquire both DWI and functional data (e.g., fMRI or BOLD) from the same subjects? If this data exists, how well do the two datasets correlate?

Line 68- Which methodologies are not rigorously applied?

Line 236- What is meant by 'degree'?

Suggestions

1) Insert a few bullet points at the end to communicate the main points more efficiently. Could include these as a subsection in the conclusion, or a table or figure. 

2) Describe the specific brain regions that were under investigation in the various reports cited. 

3) Comment on the disparity or similarity between structure/function correlation in specific brain regions. 

4) Add the word brain to the title and keywords to make the work easier for people to find.

5)The paper does not read well, particularly the introduction. For example, lines 69-71 are repeated almost verbatim in lines 77-80. The phrase 'the methods and methodologies are not always rigorously founded when they are applied' is used twice (lines 15-16 and lines 68-69) but not explained in either case. Grammar is needed to improve.

Author Response

(The authors gave the same response as above.)
